# Deep Poisson gamma dynamical systems

**Dandan Guo,**     **Bo Chen,**[*]     **Hao Zhang**
National Laboratory of Radar Signal Processing
Collaborative Innovation Center of Information Sensing and Understanding
Xidian University, Xi'an, China
gdd_xidian@126.com, bchen@mail.xidian.edu.cn, zhanghao_xidian@163.com

**Mingyuan Zhou**
McCombs School of Business
The University of Texas at Austin
Austin, TX 78712, USA
mingyuan.zhou@mccombs.utexas.edu

## Abstract

We develop deep Poisson-gamma dynamical systems (DPGDS) to model sequentially observed multivariate count data, improving previously proposed models by not only mining deep hierarchical latent structure from the data, but also capturing both first-order and long-range temporal dependencies. Using sophisticated but simple-to-implement data augmentation techniques, we derived closed-form Gibbs sampling update equations by first backward and upward propagating auxiliary latent counts, and then forward and downward sampling latent variables. Moreover, we develop stochastic gradient MCMC inference that is scalable to very long multivariate count time series. Experiments on both synthetic and a variety of real-world data demonstrate that the proposed model not only has excellent predictive performance, but also provides highly interpretable multilayer latent structure to represent hierarchical and temporal information propagation.

## 1 Introduction

The need to model time-varying count vectors $x_1, ..., x_T$ appears in a wide variety of settings, such as text analysis, international relation study, social interaction understanding, and natural language processing [1–9]. To model these count data, it is important to not only consider the sparsity of high-dimensional data and robustness to over-dispersed temporal patterns, but also capture complex dependencies both within and across time steps. In order to move beyond linear dynamical systems (LDS) [10] and its nonlinear generalization [11] that often make the Gaussian assumption [12], the gamma process dynamic Poisson factor analysis (GP-DPFA) [5] factorizes the observed time-varying count vectors under the Poisson likelihood as $x_t \sim \text{Poisson}(\Phi\theta_t)$, and transmit temporal information smoothly by evolving the factor scores with a gamma Markov chain as $\theta_t \sim \text{Gamma}(\theta_{t-1}, \beta)$, which has highly desired strong non-linearity. To further capture cross-factor temporal dependence, a transition matrix $\Pi$ is further used in Poisson–gamma dynamical system (PGDS) [7] as $\theta_t \sim \text{Gamma}(\Pi\theta_{t-1}, \beta)$. However, these shallow models may still have shortcomings in capturing long-range temporal dependencies [8]. For example, if given $\theta_t$, then $\theta_{t+1}$ no longer depends on $\theta_{t-k}$ for all $k \geq 1$.

Deep probabilistic models are widely used to capture the relationships between latent variables across multiple stochastic layers [4, 8, 13–16]. For example, deep dynamic Poisson factor analysis (DDPFA)

---

[*]Corresponding author

[8] utilizes recurrent neural networks (RNN) [3] to capture long-range temporal dependencies of the factor scores. The latent variables and RNN parameters, however, are separately inferred. Deep temporal sigmoid belief network (DTSBN) [4] is a deep dynamic generative model defined as a sequential stack of sigmoid belief networks (SBNs), whose hidden units are typically restricted to be binary. Although a deep structure is designed to describe complex long-range temporal dependencies, how the layers in DTSBN are related to each other lacks an intuitive interpretation, which is of paramount interest for a multilayer probabilistic model [15].

In this paper, we present deep Poisson gamma dynamical systems (DPGDS), a deep probabilistic dynamical model that takes the advantage of the hierarchical structure to efficiently incorporate both between-layer and temporal dependencies, while providing rich interpretation. Moving beyond DTSBN using binary hidden units, we build a deep dynamic directed network with gamma distributed nonnegative real hidden units, inferring a multilayer contextual representation of multivariate time-varying count vectors. Consequently, DPGDS can handle highly overdispersed counts, capturing the correlations between the visible/hidden features across layers and over times using the gamma belief network [15]. Combing the deep and temporal structures shown in Fig. 1(a), DPGDS breaks the assumption that given $\boldsymbol{\theta}_t$, $\boldsymbol{\theta}_{t+1}$ no longer depends on $\boldsymbol{\theta}_{t-k}$ for $k \geq 1$, suggesting that it may better capture long-range temporal dependencies. As a result, the model can allow more specific information, which are also more likely to exhibit fast temporal changing, to transmit through lower layers, while allowing more general information, which are also more likely to slowly evolve over time, to transmit through higher layers. For example, as shown in Fig. 1(b) that is learned from GDELT2003 with DPGDS, when analyzing these international events, the factors at lower layers are more specific to discover the relationships between the different countries, whereas those at higher layers are more general to reflect the conflicts between the different areas consisting of several related countries, or the ones occurring simultaneously, and the latent representation $\boldsymbol{\theta}_t$ at a lower layer varies more intensely than that at a higher layer.

Distinct from DDPFA [8] that adopts a two-stage inference, the latent variables of DPGDS can be jointly trained with both a Backward-Upward–Forward-Downward (BUFD) Gibbs sampler and a sophisticated stochastic gradient MCMC (SGMCMC) algorithm that is scalable to very long multivariate time series [17–21]. Furthermore, the factors learned at each layer can refine the understanding and analysis of sequentially observed multivariate count data, which, to the best of our knowledge, may be very challenging for existing methods. Finally, based on a diverse range of real-world data sets, we show that DPGDS exhibits excellent predictive performance, inferring interpretable latent structure with well captured long-range temporal dependencies.

## 2 Deep Poisson gamma dynamic systems

Shown in Fig. 1(a) is the graphical representation of a three-hidden-layer DPGDS. Let us denote $\theta \sim \text{Gam}(a, c)$ as a gamma random variable with mean $a/c$ and variance $a/c^2$. Given a set of $V$-dimensional sequentially observed multivariate count vectors $\boldsymbol{x}_1, ..., \boldsymbol{x}_T$, represented as a $V \times T$ matrix $\mathbf{X}$, the generative process of a $L$-hidden-layer DPGDS, from top to bottom, is expressed as

$$\boldsymbol{\theta}_t^{(L)} \sim \text{Gam}\left(\tau_0 \boldsymbol{\Pi}^{(L)} \boldsymbol{\theta}_{t-1}^{(L)}, \tau_0\right), \cdots, \boldsymbol{\theta}_t^{(l)} \sim \text{Gam}\left(\tau_0 (\boldsymbol{\Phi}^{(l+1)} \boldsymbol{\theta}_t^{(l+1)} + \boldsymbol{\Pi}^{(l)} \boldsymbol{\theta}_{t-1}^{(l)}), \tau_0\right), \cdots,$$

$$\boldsymbol{\theta}_t^{(1)} \sim \text{Gam}\left(\tau_0 (\boldsymbol{\Phi}^{(2)} \boldsymbol{\theta}_t^{(2)} + \boldsymbol{\Pi}^{(1)} \boldsymbol{\theta}_{t-1}^{(1)}), \tau_0\right), \quad \boldsymbol{x}_t^{(1)} \sim \text{Pois}\left(\delta_t^{(1)} \boldsymbol{\Phi}^{(1)} \boldsymbol{\theta}_t^{(1)}\right), \tag{1}$$

where $\boldsymbol{\Phi}^{(l)} \in \mathbb{R}_+^{K_{l-1} \times K_l}$ is the factor loading matrix at layer $l$, $\boldsymbol{\theta}_t^{(l)} \in \mathbb{R}_+^{K_l}$ the hidden units of layer $l$ at time $t$, and $\boldsymbol{\Pi}^{(l)} \in \mathbb{R}_+^{K_l \times K_l}$ a transition matrix of layer $l$ that captures cross-factor temporal dependencies. We denote $\delta_t^{(1)} \in \mathbb{R}_+$ as a scaling factor, reflecting the scale of the counts at time $t$; one may also set $\delta_t^{(1)} = \delta^{(1)}$ for $t = 1, ..., T$. We denote $\tau_0 \in \mathbb{R}_+$ as a scaling hyperparameter that controls the temporal variation of the hidden units. The multilayer time-varying hidden units $\boldsymbol{\theta}_t^{(l)}$ are well suited for downstream analysis, as will be shown below.

DPGDS factorizes the count observation $\boldsymbol{x}_t^{(1)}$ into the product of $\delta_t^{(1)}$, $\boldsymbol{\Phi}^{(1)}$, and $\boldsymbol{\theta}_t^{(1)}$ under the Poisson likelihood. It further factorizes the shape parameters of the gamma distributed $\boldsymbol{\theta}_t^{(l)}$ of layer $l$ at time $t$ into the sum of $\boldsymbol{\Phi}^{(l+1)} \boldsymbol{\theta}_t^{(l+1)}$, capturing the dependence between different layers, and $\boldsymbol{\Pi}^{(l)} \boldsymbol{\theta}_{t-1}^{(l)}$, capturing the temporal dependence at the same layer. At the top layer, $\boldsymbol{\theta}_t^{(L)}$ is only

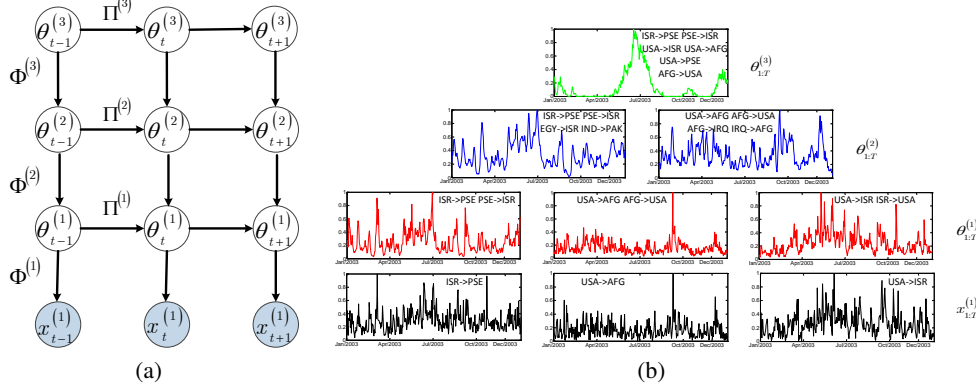

Figure 1: Graphical model and illustration for a three-hidden-layer deep Poisson Gamma Dynamical System (DPGDS). (a) The generative model; (b) Visualization of data and latent factors learned from GDELT2003, with the black, red, blue and green lines denoting the observed data, temporal trajectories of example latent factors at layer 1, 2, 3, respectively.

dependent on $\mathbf{\Pi}^{(L)}\boldsymbol{\theta}_{t-1}^{(L)}$, and at $t=1$, $\boldsymbol{\theta}_1^{(l)} \sim \text{Gam}\left(\tau_0 \mathbf{\Phi}^{(l+1)}\boldsymbol{\theta}_1^{(l+1)}, \tau_0\right)$ for $l = 1, \ldots, L-1$ and $\boldsymbol{\theta}_1^{(L)} \sim \text{Gam}\left(\tau_0 \nu_k^{(L)}, \tau_0\right)$. To complete the hierarchical model, we introduce $K_l$ factor weights $\boldsymbol{\nu}^{(l)} = (\nu_1^{(l)}, ..., \nu_{K_l}^{(l)})$ in layer $l$ to model the strength of each factor, and for $l = 1, ..., L$, we let

$$\boldsymbol{\pi}_k^{(l)} \sim \text{Dir}(\nu_1^{(l)}\nu_k^{(l)}, ..., \nu_{k-1}^{(l)}\nu_k^{(l)}, \xi^{(l)}\nu_k^{(l)}, \nu_{k+1}^{(l)}\nu_k^{(l)} ..., \nu_{K_l}^{(l)}\nu_k^{(l)}), \quad \nu_k^{(l)} \sim \text{Gam}(\tfrac{\gamma_0}{K_l}, \beta^{(l)}). \quad (2)$$

Note that $\boldsymbol{\pi}_k^{(l)}$ is the $k^{th}$ column of $\mathbf{\Pi}^{(l)}$ and $\pi_{k_1 k_2}^{(l)}$ can be interpreted as the probability of transiting from topic $k_2$ of the previous time to topic $k_1$ of the current time at layer $l$.

Finally, we place Dirichlet priors on the factor loadings and draw other parameters from a noninformative gamma prior: $\boldsymbol{\phi}_k^{(l)} = (\phi_{1k}^{(l)}, ..., \phi_{K_{l-1}k}^{(l)}) \sim \text{Dir}(\eta^{(l)}, ..., \eta^{(l)})$, and $\delta_t^{(1)}, \xi^{(l)}, \beta^{(l)} \sim \text{Gam}(\epsilon_0, \epsilon_0)$. Note that imposing Dirichlet distributions on the columns of $\mathbf{\Pi}^{(l)}$ and $\mathbf{\Phi}^{(l)}$ not only makes the latent representation more identifiable and interpretable, but also facilitates inference, as will be shown in the next section. Clearly when $L = 1$, DPGDS reduces to PGDS [7]. In real-world applications, a binary observation can be linked to a latent count using the Bernoulli-Poisson link as $b = 1(n \geq 1), n \sim \text{Pois}(\lambda)$ [22]. Nonnegative-real-valued matrix can also be linked to a latent count matrix via a Poisson randomized gamma distribution as $x \sim \text{Gam}(n, c), n \sim \text{Pois}(\lambda)$ [23].

**Hierarchical structure:** To interpret the hierarchical structure of (1), we notice that $\mathbb{E}\left[\boldsymbol{x}_t^{(1)} \,|\, \boldsymbol{\theta}_t^{(l)}, \{\mathbf{\Phi}^{(p)}\}_{p=1}^l\right] = \left[\prod_{p=1}^l \mathbf{\Phi}^{(p)}\right] \boldsymbol{\theta}_t^{(l)}$ if the temporal structure is ignored. Thus it is straightforward to interpret $\boldsymbol{\phi}_k^{(l)}$ by projecting them to the bottom data layer as $\left[\prod_{t=1}^{l-1} \mathbf{\Phi}^{(t)}\right] \boldsymbol{\phi}_k^{(l)}$, which are often quite specific at the bottom layer and become increasingly more general when moving upwards, as will be shown below in Fig. 5(a).

**Long-range temporal dependencies**: Using the law of total expectations on (1), for a three-hidden-layer DPGDS shown in Fig. 1(a), we have

$$\mathbb{E}[\boldsymbol{x}_t^{(1)} \,|\, \boldsymbol{\theta}_{t-1}^{(1)}, \boldsymbol{\theta}_{t-2}^{(2)}, \boldsymbol{\theta}_{t-3}^{(3)}]/\delta_t^{(1)} = \mathbf{\Phi}^{(1)}\mathbf{\Pi}^{(1)}\boldsymbol{\theta}_{t-1}^{(1)} + \mathbf{\Phi}^{(1)}\mathbf{\Phi}^{(2)}[\mathbf{\Pi}^{(2)}]^2\boldsymbol{\theta}_{t-2}^{(2)}$$
$$+ \mathbf{\Phi}^{(1)}\mathbf{\Phi}^{(2)}(\mathbf{\Pi}^{(2)}\mathbf{\Phi}^{(3)} + \mathbf{\Phi}^{(3)}\mathbf{\Pi}^{(3)})[\mathbf{\Pi}^{(3)}]^2\boldsymbol{\theta}_{t-3}^{(3)}, \quad (3)$$

which suggests that $\{\mathbf{\Pi}^{(l)}\}_{l=1}^L$ play the role of transiting the latent representation across time and, different from most existing dynamic models, DPGDS can capture and transmit long-range temporal information (often general and change slowly over time) through its higher hidden layers.

## 3   Scalable MCMC inference

In this paper, in each iteration, across layers and times, we first exploit a variety of data augmentation techniques for count data to "backward" and "upward" propagate auxiliary latent counts, with which

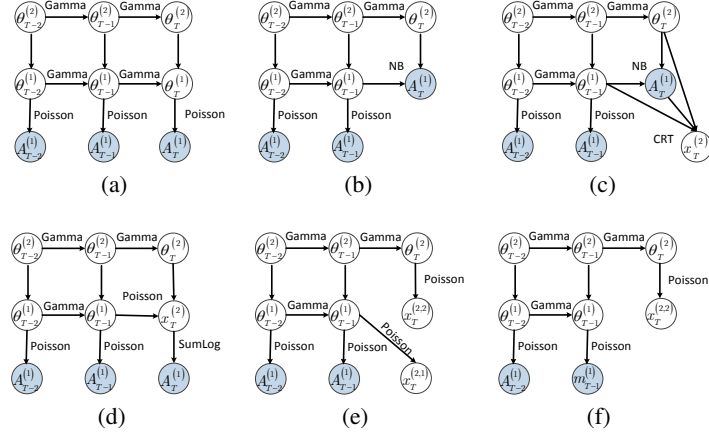

Figure 2: Graphical representation of the model and data augmentation and marginalization based inference scheme. (a) An alternative representation of layer $l = 1$ using the relationships between the Poisson and multinomial distributions; (b) A negative binomial distribution based representation that marginalizes out the gamma from the Poisson distributions, corresponding to (4) for $t = T$; (c) An equivalent representation that introduces CRT distributed auxiliary variables, corresponding to (5); (d) An equivalent representation using **P3**, corresponding to (6); (e) An equivalent representation obtained by using **P1**, corresponding to (7); (f) A representation obtained by repeating the same augmentation-marginalization steps described in (a).

we then "downward" and "forward" sample latent variables, leading to a Backward-Upward–Forward-Downward Gibbs sampling (BUFD) Gibbs sampling algorithm.

## 3.1  Backward and upward propagation of latent counts

Different from PGDS that has only backward propagation for latent counts, DPGDS have both backward and upward ones due to its deep hierarchical structure. To derive closed-form Gibbs sampling update equations, we exploit three useful properties for count data, denoted as **P1**, **P2**, and **P3** [7, 24], respectively, as presented in the Appendix. Let us denote $x \sim \mathrm{NB}(r, p)$ as the negative binomial distribution with probability mass function $P(x = k) = \frac{\Gamma(k+r)}{k!\Gamma(r)}p^k(1-p)^r$, where $k \in \{0, 1, \dots\}$. First, we can augment each count $x_{vt}^{(1)}$ in (1) into the summation of $K_1$ latent counts that are smaller or equal as $x_{vt}^{(1)} = \sum_{k=1}^{K_1} A_{vkt}^{(1)}$, $A_{vkt}^{(1)} \sim \mathrm{Pois}(\delta_t^{(1)}\phi_{vk}^{(1)}\theta_{kt}^{(1)})$, with $A_{\cdot kt}^{(1)} = \sum_{v=1}^{V} A_{vkt}^{(1)}$. Since $\sum_{v=1}^{V} \phi_{vk}^{(1)} = 1$ by construction, we also have $A_{\cdot kt}^{(1)} \sim \mathrm{Pois}(\delta_t^{(1)}\theta_{kt}^{(1)})$, as shown in Fig. 2(a). We start with $\boldsymbol{\theta}_T^{(1)}$ at the last time point $T$, as none of the other time-step factors depend on it in their priors. Via **P2**, as shown in Fig. 2(b), we can marginalize out $\theta_{kT}^{(1)}$ to obtain

$$A_{\cdot kT}^{(1)} \sim \mathrm{NB}\left[\tau_0 \left(\sum_{k_2=1}^{K_2} \phi_{kk_2}^{(2)}\theta_{k_2T}^{(2)} + \sum_{k_1=1}^{K_1} \pi_{kk_1}^{(1)}\theta_{k_1,T-1}^{(1)}\right), g(\zeta_T^{(1)})\right], \qquad (4)$$

where $\zeta_T^{(1)} = \ln(1 + \frac{\delta_T^{(1)}}{\tau_0})$ and $g(\zeta) = 1 - \exp(-\zeta)$.

In order to marginalize out $\boldsymbol{\theta}_{T-1}^{(1)}$, as shown in Fig. 2(c), we introduce an auxiliary variable following the Chines restaurant table (CRT) distribution [24] as

$$x_{kT}^{(2)} \sim \mathrm{CRT}\left[A_{\cdot kT}^{(1)}, \tau_0 \left(\sum_{k_2=1}^{K_2} \phi_{kk_2}^{(2)}\theta_{k_2T}^{(2)} + \sum_{k_1=1}^{K_1} \pi_{kk_1}^{(1)}\theta_{k_1,T-1}^{(1)}\right)\right]. \qquad (5)$$

As shown in Fig. 2(d), we re-express the joint distribution over $A_{\cdot kT}^{(1)}$ and $x_{kT}^{(2)}$ according to **P3** as

$$A_{\cdot kT}^{(1)} \sim \mathrm{SumLog}(x_{kT}^{(2)}, g(\zeta_T^{(1)})), \quad x_{kT}^{(2)} \sim \mathrm{Pois}\left[\zeta_T^{(1)}\tau_0 \left(\sum_{k_2=1}^{K_2} \phi_{kk_2}^{(2)}\theta_{k_2T}^{(2)} + \sum_{k_1=1}^{K_1} \pi_{kk_1}^{(1)}\theta_{k_1,T-1}^{(1)}\right)\right], \quad (6)$$

where the sum-logarithmic (SumLog) distribution is defined as in Zhou and Carin [24]. Via **P1**, as in Fig. 2(e), the Poisson random variable $x_{kT}^{(2)}$ in (6) can be augmented as $x_{kT}^{(2)} = x_{kT}^{(2,1)} + x_{kT}^{(2,2)}$, where

$$x_{kT}^{(2,1)} \sim \text{Pois}(\zeta_T^{(1)} \tau_0 \sum_{k_1=1}^{K_1} \pi_{kk_1}^{(1)} \theta_{k_1,T-1}^{(1)}), \quad x_{kT}^{(2,2)} \sim \text{Pois}(\zeta_T^{(1)} \tau_0 \sum_{k_2=1}^{K_2} \phi_{kk_2}^{(2)} \theta_{k_2 T}^{(2)}). \quad (7)$$

It is obvious that due to the deep dynamic structure, the count at layer two $x_{kT}^{(2)}$ is divided into two parts: one from time $T-1$ at layer one, while the other from time $T$ at layer two. Furthermore, $\zeta_T^{(1)}$ is the scaling factor at layer two, which is propagated from the one at layer one $\delta_T^{(1)}$. Repeating the process all the way back to $t = 1$, and from $l = 1$ up to $l = L$, we are able to marginalize out all gamma latent variables $\{\mathbf{\Theta}\}_{t=1,l=1}^{T,L}$ and provide closed-form conditional posteriors for all of them.

## 3.2 Backward-upward–forward-downward Gibbs sampling

**Sampling auxiliary counts:** This step is about the "backward" and "upward" pass. Let us denote $Z_{\cdot kt}^{(l)} = \sum_{k_l=1}^{K_l} Z_{k_l kt}^{(l)}$, $Z_{\cdot k,T+1}^{(l)} = 0$, and $x_{kt}^{(1,1)} = x_{vt}^{(1)}$. Working backward for $t = T, ..., 2$ and upward for $l = 1, ..., L$, we draw

$$(A_{k1t}^{(l)}, ..., A_{kK_l t}^{(l)}) \sim \text{Multi}\left(x_{kt}^{(l,l)}; \frac{\phi_{k1}^{(l)} \theta_{1t}^{(l)}}{\sum_{k_l=1}^{K_l} \phi_{kk_l}^{(l)} \theta_{k_l t}^{(l)}}, ..., \frac{\phi_{kK_l}^{(l)} \theta_{K_l t}^{(l)}}{\sum_{k_l=1}^{K_l} \phi_{kk_l}^{(l)} \theta_{k_l t}^{(l)}}\right), \quad (8)$$

$$x_{kt}^{(l+1)} \sim \text{CRT}\left[A_{\cdot kt}^{(l)} + Z_{\cdot k,t+1}^{(l)}, \tau_0 \left(\sum_{k_{l+1}=1}^{K_{l+1}} \phi_{kk_{l+1}}^{(l+1)} \theta_{k_{l+1}t}^{(l+1)} + \sum_{k_l=1}^{K_l} \pi_{kk_1}^{(l)} \theta_{k_1,t-1}^{(l)}\right)\right]. \quad (9)$$

Note that via the deep structure, the latent counts $x_{kt}^{(l+1)}$ will be influenced by the effects from both of time $t-1$ at layer $l$ and time $t$ at layer $l+1$. With $p_1 := \sum_{k_l=1}^{K_l} \pi_{kk_l}^{(l)} \theta_{k_l,t-1}^{(l)}$ and $p_2 := \sum_{k_{l+1}=1}^{K_{l+1}} \phi_{kk_{l+1}}^{(l+1)} \theta_{k_{l+1}t}^{(l+1)}$, we can sample the latent counts at layer $l$ and $l+1$ by

$$(x_{kt}^{(l+1,l)}, x_{kt}^{(l+1,l+1)}) \sim \text{Multi}\left(x_{kt}^{(l+1)}, p_1/(p_1+p_2), p_2/(p_1+p_2)\right), \quad (10)$$

and then draw

$$(Z_{k1t}^{(l)}, ..., Z_{kK_l t}^{(l)}) \sim \text{Multi}\left(x_{kt}^{(l+1,l)}; \frac{\pi_{k1}^{(l)} \theta_{1,t-1}^{(l)}}{\sum_{k_l=1}^{K_l} \pi_{kk_l}^{(l)} \theta_{k_l,t-1}^{(l)}}, ..., \frac{\pi_{kK_l}^{(l)} \theta_{K_l,t-1}^{(l)}}{\sum_{k_l=1}^{K_l} \pi_{kk_l}^{(l)} \theta_{k_l,t-1}^{(l)}}\right). \quad (11)$$

**Sampling hidden units $\theta_t^{(l)}$ and calculating $\zeta_t^{(l)}$:** Given the augmented latent count variables, working forward for $t = 1, ..., T$ and downward for $l = L, ..., 1$, we can sample

$$\theta_{kt}^{(l)} \sim \text{Gamma}\left[A_{\cdot kt}^{(l)} + Z_{\cdot k(t+1)}^{(l)} + \tau_0 \left(\sum_{k_{l+1}=1}^{K_{l+1}} \phi_{kk_{l+1}}^{(l+1)} \theta_{k_{l+1}t}^{(l+1)} + \sum_{k_l=1}^{K_l} \pi_{kk_l}^{(l)} \theta_{k_2,t-1}^{(l)}\right), \\ \tau_0 \left(1 + \zeta_t^{(l-1)} + \zeta_{t+1}^{(l)}\right)\right], \quad (12)$$

where $\zeta_t^{(0)} = \frac{\delta_t^{(1)}}{\tau_0}$ and $\zeta_t^{(l)} = \ln\left(1 + \zeta_t^{(l-1)} + \zeta_{t+1}^{(l)}\right)$. Note if $\delta_t^{(1)} = \delta^{(1)}$ for $t = 1, ..., T$, then we may let $\zeta^{(l)} = -\mathbf{W}_{-1}(-\exp(-1 - \zeta^{(l-1)})) - 1 - \zeta^{(l-1)}$, where the function $\mathbf{W}_{-1}$ is the lower real part of the Lambert W function [7, 25]. From (12), we can find that the conditional posterior of $\boldsymbol{\theta}_t^{(l)}$ is parameterized by not only both $\mathbf{\Phi}^{(l+1)} \boldsymbol{\theta}_t^{(l+1)}$ and $\mathbf{\Pi}^{(l)} \boldsymbol{\theta}_{t-1}^{(l)}$, which represent the information from layer $l+1$ (downward) and time $t-1$ (forward), respectively, but also both $A_{\cdot,:,t}^{(l)}$ and $Z_{\cdot,:,t+1}^{(l)}$, which record the message from layer $l-1$ (upward) in (8) and time $t+1$ (backward) in (11), respectively. We describe the BUFD Gibbs sampling algorithm for DPGDS in Algorithm 1 and provide more details in the Appendix.

## 3.3 Stochastic gradient MCMC inference

Although the proposed BUFD Gibbs sampling algorithm for DPGDS has closed-form update equations, it requires processing all time-varying vectors at each iteration and hence has limited scalability [26]. To allow for scalable inference, we apply the topic-layer-adaptive stochastic gradient

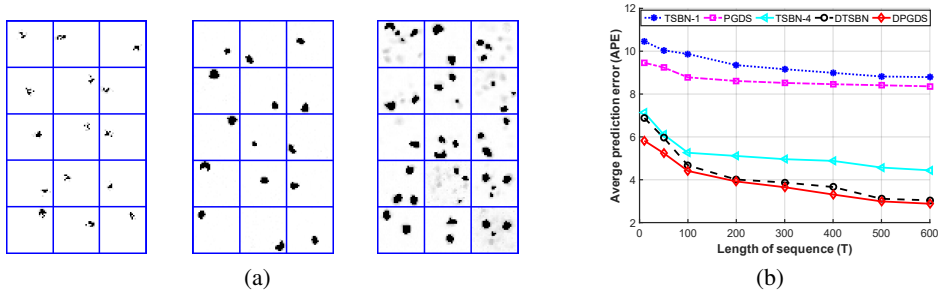

<center>(a)                    (b)</center>

Figure 3: Results on the bouncing ball data set. (a) Shown in the first to third columns are the top fifteen latent factors learned by a three-hidden-layer DPGDS at layers 1, 2, and 3, respectively; (b) The average prediction errors as a function of the sequence length for various algorithms.

Riemannian (TLASGR) MCMC algorithm described in Cong et al. [27] and Zhang et al. [26], which can be used to sample simplex-constrained global parameters [28] in a mini-batch based manner. It improves its sampling efficiency via the use of the Fisher information matrix (FIM) [29], with adaptive step-sizes for the latent factors and transition matrices of different layers. More specifically, for $\boldsymbol{\pi}_k^{(l)}$, column $k$ of the transition matrix $\boldsymbol{\Pi}^{(l)}$ of layer $l$, its sampling can be efficiently realized as

$$
\left(\boldsymbol{\pi}_k^{(l)}\right)_{n+1} = \Bigg[\left(\boldsymbol{\pi}_k^{(l)}\right)_n + \frac{\varepsilon_n}{M_k^{(l)}}\Big[\left(\rho\tilde{\boldsymbol{z}}_{:k\cdot}^{(l)} + \boldsymbol{\eta}_{:k}^{(l)}\right) - \left(\rho\tilde{z}_{\cdot k\cdot}^{(l)} + \eta_{\cdot k}^{(l)}\right)\left(\boldsymbol{\pi}_k^{(l)}\right)_n\Big]
$$
$$
+ \mathcal{N}\left(0, \frac{2\varepsilon_n}{M_k^{(l)}}\left[\mathrm{diag}(\boldsymbol{\pi}_k^{(l)})_n - (\boldsymbol{\pi}_k^{(l)})_n(\boldsymbol{\pi}_k^{(l)})_n^T\right]\right)\Bigg]_{\angle}, \tag{13}
$$

where $M_k^{(l)}$ is calculated using the estimated FIM, both $\tilde{\boldsymbol{z}}_{:k\cdot}^{(l)}$ and $\tilde{z}_{\cdot k\cdot}^{(l)}$ come from the augmented latent counts $Z^{(l)}$, $[.]_{\angle}$ denotes a simplex constraint, and $\boldsymbol{\eta}_{:k}^{(l)}$ denotes the prior of $\boldsymbol{\pi}_k^{(l)}$. The update of $\boldsymbol{\Phi}^{(l)}$ is the same with Cong et al. [27], and all the other global parameters are sampled using SGNHT [20]. We provide the details of the SGMCMC for DPGDS in Algorithm 2 in the Appendix.

## 4 Experiments

In this section, we present experimental results on a synthetic dataset and five real-world datasets. For a fair comparison, we consider PGDS [7], GP-DPFA [5], DTSBN [4], and GPDM [11] that can be considered as a dynamic generalization of the Gaussian process latent variable model of Lawrence [30], using the code provided by the authors. Note that as shown Schein et al. [7] and Gan et al. [4], PGDS and DTSBN are state-of-the-art count time series modeling algorithms that outperform a wide variety of previously proposed ones, such as LDS [12] and DRFM [31]. The hyperparameter settings of PGDS, GP-DPFA, GPDM, TSBN, and DTSBN are the same as their original settings [4, 5, 7, 11]. For DPGDS, we set $\tau_0 = 1, \gamma_0 = 100, \eta_0 = 0.1$ and $\epsilon_0 = 0.1$. We use $[K^{(1)}, K^{(2)}, K^{(3)}] = [200, 100, 50]$ for both DPGDS and DTSBN and $K = 200$ for PGDS, GP-DPFA, GPDM, and TSBN. For PGDS, GP-DPFA, GPDM, and DPGDS, we run 2000 Gibbs sampling as burn-in and collect 3000 samples for evaluation. We also use SGMCMC to infer DPGDS, with 5000 collection samples after 5000 burn-in steps, and use 10000 SGMCMC iterations for both TSBN and DTSBN to evaluate their performance.

### 4.1 Synthetic dataset

Following the literature [1, 4], we consider sequences of different lengths, including $T = 10, 50, 100, 200, 300, 400, 500$ and $600$, and generate 50 synthetic bouncing ball videos for training, and 30 ones for testing. Each video frame is a binary-valued image with size $30 \times 30$, describing the location of three balls within the image. Both TSBN and DTSBN model it with the Bernoulli likelihood, while both PGDS and DPGDS use the Bernoulli-Poisson link [22].

As shown in Fig. 3(b), the average prediction errors of all algorithms decrease as the training sequence length increases. A higher-order TSBN, TSBN-4, performs much better than the first-order TSBN

Table 1: Top-$M$ results on real-world text data

| Model | Top-$M$ | GDELT ($T=365$) | ICEWS ($T=365$) | SOTU ($T=225$) | DBLP ($T=14$) | NIPS ($T=17$) |
|---|---|---|---|---|---|---|
| GPDPFA | MP | $0.611_{\pm0.001}$ | $0.607_{\pm0.002}$ | $0.379_{\pm0.002}$ | $\mathbf{0.435}_{\pm0.009}$ | $0.843_{\pm0.005}$ |
| | MR | $0.145_{\pm0.002}$ | $0.235_{\pm0.005}$ | $0.369_{\pm0.002}$ | $0.254_{\pm0.005}$ | $0.050_{\pm0.001}$ |
| | PP | $0.447_{\pm0.014}$ | $0.465_{\pm0.008}$ | $0.617_{\pm0.013}$ | $0.581_{\pm0.011}$ | $0.807_{\pm0.006}$ |
| PGDS | MP | $0.679_{\pm0.001}$ | $0.658_{\pm0.001}$ | $0.375_{\pm0.002}$ | $0.419_{\pm0.004}$ | $0.864_{\pm0.004}$ |
| | MR | $\mathbf{0.150}_{\pm0.001}$ | $\mathbf{0.245}_{\pm0.001}$ | $0.373_{\pm0.002}$ | $0.252_{\pm0.002}$ | $0.050_{\pm0.001}$ |
| | PP | $0.420_{\pm0.017}$ | $0.455_{\pm0.008}$ | $0.612_{\pm0.018}$ | $0.566_{\pm0.008}$ | $0.802_{\pm0.020}$ |
| GPDM | MP | $0.520_{\pm0.001}$ | $0.530_{\pm0.002}$ | $0.274_{\pm0.001}$ | $0.388_{\pm0.004}$ | $0.355_{\pm0.008}$ |
| | MR | $0.141_{\pm0.001}$ | $0.234_{\pm0.001}$ | $0.261_{\pm0.002}$ | $0.146_{\pm0.005}$ | $0.050_{\pm0.001}$ |
| | PP | $0.362_{\pm0.021}$ | $0.185_{\pm0.017}$ | $0.587_{\pm0.016}$ | $0.509_{\pm0.008}$ | $0.384_{\pm0.028}$ |
| TSBN | MP | $0.594_{\pm0.007}$ | $0.471_{\pm0.001}$ | $0.360_{\pm0.001}$ | $0.403_{\pm0.012}$ | $0.788_{\pm0.005}$ |
| | MR | $0.124_{\pm0.001}$ | $0.158_{\pm0.001}$ | $0.275_{\pm0.001}$ | $0.194_{\pm0.001}$ | $0.050_{\pm0.001}$ |
| | PP | $0.418_{\pm0.019}$ | $0.445_{\pm0.031}$ | $0.611_{\pm0.001}$ | $0.527_{\pm0.003}$ | $0.692_{\pm0.017}$ |
| DTSBN-2 | MP | $0.439_{\pm0.001}$ | $0.475_{\pm0.002}$ | $0.370_{\pm0.004}$ | $0.407_{\pm0.003}$ | $0.756_{\pm0.001}$ |
| | MR | $0.134_{\pm0.001}$ | $0.208_{\pm0.001}$ | $0.361_{\pm0.001}$ | $0.248_{\pm0.007}$ | $0.050_{\pm0.001}$ |
| | PP | $0.391_{\pm0.001}$ | $0.446_{\pm0.001}$ | $0.587_{\pm0.027}$ | $0.522_{\pm0.005}$ | $0.737_{\pm0.004}$ |
| DTSBN-3 | MP | $0.411_{\pm0.001}$ | $0.431_{\pm0.001}$ | $\mathbf{0.450}_{\pm0.008}$ | $0.390_{\pm0.002}$ | $0.774_{\pm0.002}$ |
| | MR | $0.141_{\pm0.001}$ | $0.189_{\pm0.001}$ | $0.274_{\pm0.001}$ | $0.252_{\pm0.004}$ | $0.050_{\pm0.001}$ |
| | PP | $0.367_{\pm0.011}$ | $0.451_{\pm0.026}$ | $0.548_{\pm0.013}$ | $0.510_{\pm0.006}$ | $0.715_{\pm0.009}$ |
| DPGDS-2 | MP | $0.688_{\pm0.002}$ | $0.659_{\pm0.001}$ | $0.379_{\pm0.002}$ | $0.430_{\pm0.009}$ | $0.867_{\pm0.008}$ |
| | MR | $0.149_{\pm0.001}$ | $0.242_{\pm0.001}$ | $0.373_{\pm0.001}$ | $0.254_{\pm0.005}$ | $0.050_{\pm0.001}$ |
| | PP | $0.443_{\pm0.025}$ | $0.473_{\pm0.012}$ | $0.622_{\pm0.014}$ | $0.582_{\pm0.007}$ | $0.814_{\pm0.035}$ |
| DPGDS-3 | MP | $\mathbf{0.689}_{\pm0.002}$ | $0.660_{\pm0.001}$ | $0.380_{\pm0.001}$ | $0.431_{\pm0.012}$ | $\mathbf{0.887}_{\pm0.002}$ |
| | MR | $\mathbf{0.150}_{\pm0.001}$ | $0.244_{\pm0.003}$ | $\mathbf{0.374}_{\pm0.002}$ | $\mathbf{0.255}_{\pm0.004}$ | $0.050_{\pm0.001}$ |
| | PP | $\mathbf{0.456}_{\pm0.015}$ | $\mathbf{0.478}_{\pm0.024}$ | $\mathbf{0.628}_{\pm0.021}$ | $\mathbf{0.600}_{\pm0.001}$ | $\mathbf{0.839}_{\pm0.007}$ |

does, suggesting that using high-order messages can help TSBN better pass useful information. As discussed above, since a deep structure provides a natural way to propagate high-order information for prediction, it is not surprising to find that both DTSBN and DPGDS, which are both multi-layer models, have exhibited superior performance. Moreover, it is clear that the proposed DPGDS consistently outperforms DTSBN under all settings.

Another advantage of DPGDS is that its inferred deep latent structure often has meaningful interpretation. As shown in Fig. 3(a), for the bouncing ball data, the inferred factors at layer one represent points or pixels, those at layer two cover larger spatial contiguous regions, some of which exhibit the shape of a single bouncing ball, and those at layer three are able to capture multiple bouncing balls. In addition, we show in Appendix B the one-step prediction frames of different models.

## 4.2 Real-world datasets

Besides the binary-valued synthetic bouncing ball dataset, we quantitatively and qualitatively evaluate all algorithms on the following real-world datasets used in Schein et al. [7]. The State-of-the-Union (SOTU) dataset consists of the text of the annual SOTU speech transcripts from 1790 to 2014. The Global Database of Events, Language, and Tone (GDELT) and Integrated Crisis Early Warning System (ICEWS) are both datasets for international relations extracted from news corpora. Note that ICEWS consists of undirected pairs, while GDELT consists of directed pairs of countries. The NIPS corpus contains the text of every NIPS conference paper from 1987 to 2003. The DBLP corpus is a database of computer science research papers. Each of these datasets is summarized as a $V \times T$ count matrix, as shown in Tab. 1. Unless specified otherwise, we choose the top 1000 most frequently used terms to form the vocabulary, which means we set $V = 1000$ for all real-data experiments.

### 4.2.1 Quantitative comparison

For a fair and comprehensive comparison, we calculate the precision and recall at top-$M$ [4,5,31,32], which are calculated by the fraction of the top-$M$ words that match the true ranking of the words and appear in the top-$M$ ranking, respectively, with $M = 50$. We also use the Mean Precision (MP) and Mean Recall (MR) over all the years appearing in the training set to evaluate different models. As another criterion, the Predictive Precision (PP) shows the predictive precision for the final year, for which all the observations are held out. Similar as previous methods [4,5], for each corpus, the entire data of the last year is held out, and for the documents in the previous years we randomly partition the words of each document into 80% / 20% in each trial, and we conduct five random trials to report the sample mean and standard deviation. Note that to apply GPDM, we have used Anscombe transform

[33] to preprocess the count data to mitigate the mismatch between the data and model assumption. The results on all five datasets are summarized in Tab. 1, which clearly show that the proposed DPGDS has achieved the best performance on most of the evaluation criteria, and again a deep model often improves its performance by increasing its number of layers. To add more empirical study on scalability, we have also tested the efficiency of our model on a GDELT data (from 2001 to 2005, temporal granularity of 24 hrs, with a total of 1825 time points), which is not too large so that we can still run DPGDS-Gibbs and GPDM. As shown in Fig. 4, we present how various algorithms progress over time, evaluated with MP. It takes about 1000s for DTSBN and DPGDS-SGMCMC to converge, 3.5 hrs for DPGDS-Gibbs, 5 hrs for GPDM. Clearly, our DPGDS-SGMCMC is scalable and clearly outperforms both DTSBN and GPDM. We also present in Appendix C the results of DPGDS-SGMCMC on a very long time series, on which it becomes too expensive to run a batch learning algorithm.

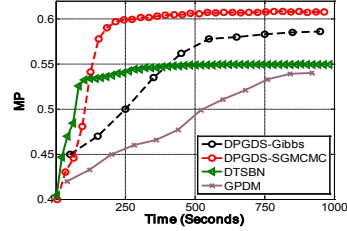

Figure 4: MP as a function of time for GDELT.

### 4.2.2  Exploratory data analysis

Compared to previously proposed dynamic systems, the proposed DPGDS, whose inferred latent structure is simple to visualize, provides much richer interpretation. More specifically, we may not only exhibit the content of each factor (topic), but also explore both the hierarchical relationships between them at different layers, and the temporal relationships between them at the same layer. Based on the results inferred on ICEWS 2001-2003 via a three hidden layer DPGDS, with the size of 200-100-50, we show in Fig. 5 how some example topics are hierarchically and temporally related to each other, and how their corresponding latent representations evolve over time.

In Fig. 5(a), we select two large-weighted topics at the top hidden layer and move down the network to include any lower-layer topics that are connected to them with sufficiently large weights. For each topic, we list all its terms whose values are larger than 1% of the largest element of the topic. It is interesting to note that topic 2 at layer three is connected to three topics at layer two, which are characterized mainly by the interactions of Israel (ISR)-Palestinian Territory (PSE), Iraq (IRQ)-USA-Iran (IRN), and North Korea (PRK)-South Korea (KOR)-USA-China (CHN)-Japan (JPN), respectively. The activation strength of one of these three interactions, known to be dominant in general during 2001-2003, can be contributed not only by a large activation of topic 2 at layer three, but also by a large activation of some other topic of the same layer (layer two) at the previous time. For example, topic 41 of layer two on "ISR-PSE, IND-PAK, RUS-UKR, GEO-RUS, AFG-PAK, SYR-USA, MNE-SRB" could be associated with the activation of topic 46 of layer two on "IND-PAK, RUS-TUR, ISR-PSE, BLR-RUS" at the previous time; and topic 99 of layer two on "PRK-KOR, JPN-USA, CHN-USA, CHN-KOR, CHN-JPN, USA-RUS" could be associated with the activation of topic 63 of layer two on "IRN-USA, CHN-USA, AUS-CHN, CHN-KOR" at the previous time.

Another instructive observation is that topic 140 of layer one on "IRQ-USA, IRQ-GBR, IRN-IRQ, IRQ-KWT, AUS-IRQ" is related not only in hierarchy to topic 34 of the higher layer on "IRQ-USA, IRQ-GBR, GBR-USA, IRQ-KWT, IRN-IRQ, SYR-USA," but also in time to topic 166 of the same layer on "ESP-USA, ESP-GBR, FRA-GBR, POR-USA," which are interactions between the member states of the North Atlantic Treaty Organization (NATO). Based on the transitions from topic 13 on "PRK-KOR" to both topic 140 on "IRQ-USA" and 77 on "ISR-PSE," we can find that the ongoing Iraq war and Israeli–Palestinian relations regain attention after the six-party talks [7].

To get an insight of the benefits attributed to the deep structure, how the latent representations of several representative topics evolve over days are shown in Fig. 5(b). It is clear that relative to these temporal factor trajectories at the bottom layer, which are specific for the bilateral interactions between two countries, these from higher layers vary more smoothly, whose corresponding high-layer topics capture the multilateral interactions between multiple closely related countries. Similar phenomena have also been demonstrated in Fig. 1(b) on GDELT2003. Moreover, we find that a spike of the temporal trajectory of topic 166 (NATO) appears right before a one of topic 140 (Iraq war), matching the above description in Fig. 5(a). Also, topic 14 of layer three and its descendants, including topic 23 of layer two and topic 48 at layer one are mainly about a breakthrough between RUS and Azerbaijan (AZE), coinciding with Putin's visit in January 2001. Additional example results for the topics and their hierarchical and temporal relationships, inferred by DPGDS on different datasets, are provided in the Appendix.

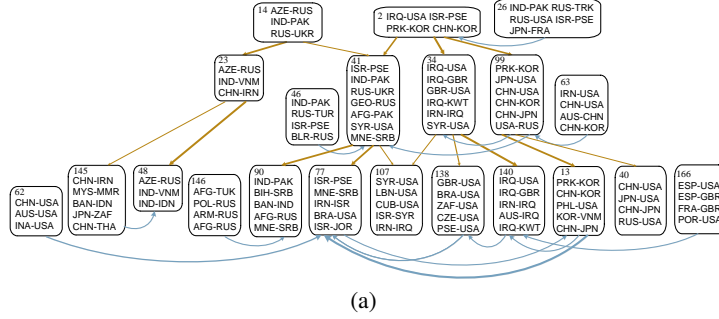

(a)

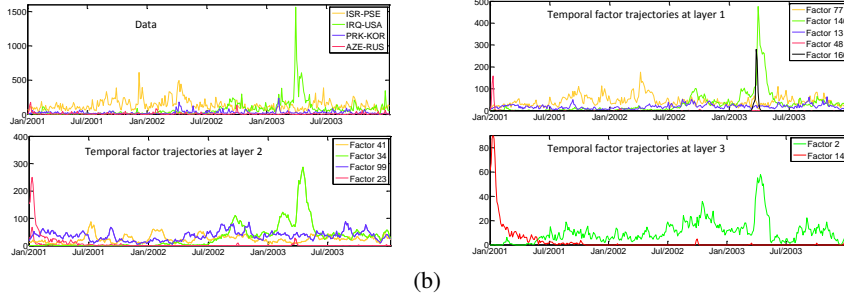

(b)

Figure 5: Topics and their temporal trajectories inferred by a three-hidden-layer DPGDS from the ICEWS 2001-2003 dataset (best viewed in color). (a) Some example topics that are hierarchically or temporally related; (b) The temporal trajectories of some inferred latent topics.

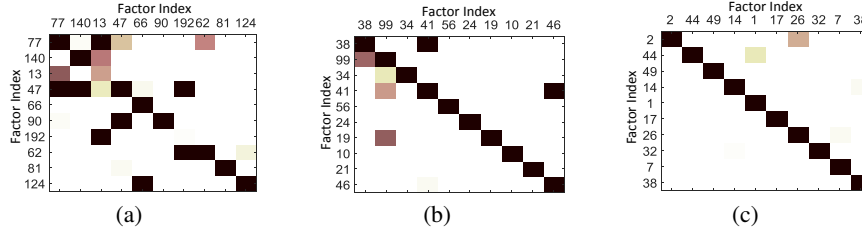

(a)　　　　　　　　(b)　　　　　　　　(c)

Figure 6: Learned transition structure on ICEWS 2001-2003 from the same DPGDS depicted in Fig. 5. Shown in (a)-(c) are transition matrices for layers 1, 2 and 3, respectively, with a darker color indicating a larger transition weight (between 0 and 1).

In Fig. 6, we also present a subset of the transition matrix $\mathbf{\Pi}^{(l)}$ in each layer, corresponding to the top ten topics, some of which have been displayed in Fig. 5(b). The transition matrix $\mathbf{\Pi}^{(l)}$ captures the cross-topic temporal dependence at layer $l$. From Fig. 6, besides the temporal transitions between the topics at the same layer, we can also see that with the increase of the layer index $l$, the transition matrix $\mathbf{\Pi}^{(l)}$ more closely approaches a diagonal matrix, meaning that the feature factors become more likely to transit to themselves, which matches the characteristic of DPGDS that the topics in higher layers have the ability to cover longer-range temporal dependencies and contain more general information, as shown in Fig. 5(a). With both the hierarchical connections between layers and dynamic transitions at the same layer, distinct from the shallow PGDS, DPGDS is equipped with a larger capacity to model diverse temporal patterns with the help of its deep structure.

## 5   Conclusions

We propose deep Poisson gamma dynamical systems (DPGDS) that take the advantage of a probabilistic deep hierarchical structure to efficiently capture both across-layer and temporal dependencies. The inferred latent structure provides rich interpretation for both hierarchical and temporal information propagation. For Bayesian inference, we develop both a Backward-Upward–Forward-Downward Gibbs sampler and a stochastic gradient MCMC (SGMCMC) that is scalable to long multivariate count/binary time series. Experimental results on a variety of datasets show that DPGDS not only exhibits excellent predictive performance, but also provides highly interpretable latent structure.

**Acknowledgements**

D. Guo, B. Chen, and H. Zhang acknowledge the support of the Program for Young Thousand Talent by Chinese Central Government, the 111 Project (No. B18039), NSFC (61771361), NSFC for Distinguished Young Scholars (61525105) and the Innovation Fund of Xidian University. M. Zhou acknowledges the support of Award IIS-1812699 from the U.S. National Science Foundation.

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
