[Supplementary Material]

# Supplementary material for deep Poisson gamma dynamical systems

Dandan Guo, Bo Chen, Hao Zhang, and Mingyuan Zhou

## A  Details of inference via Gibbs sampling for DPGDS

Inference for the DPGDS shown in (1) is challenging, as neither the conjugate prior nor closed-form maximum likelihood estimate is known for the shape parameter of a gamma distribution. Although seemingly difficult, by generalizing the data augmentation and marginalization techniques, we are able to derive a backward-upward and then forward-downward Gibbs sampling algorithm, making it simple to draw random samples to represent the posteriors of model parameters. We marginalize over $\Theta^{(1:L)}$ by performing a "ackward" and "upward" filters, starting with $\theta_T^{(1)}$. We repeatedly exploit the following three properties:

**Property 1 (P1)**: if $y_{\cdot kt} = \sum_{n=1}^{N} y_n$, where $y_n \sim \text{Pois}(\theta_n)$ are independent Poisson-distributed random variables, then $(y_1, ..., y_n) \sim \text{Multi}\left(y_\cdot, \frac{\theta_1}{\sum_{n=1}^{N}\theta_n}, ..., \frac{\theta_N}{\sum_{n=1}^{N}\theta_n}\right)$ and $y_\cdot \sim \text{Pois}(\sum_{n=1}^{N}\theta_n)$ [34, 35].

**Property 2 (P2)**: $y \sim \text{Pois}(c\theta)$, where c is a constant, and $\theta \sim \text{Gam}(a, b)$ then $y \sim \text{NB}\left(a, \frac{c}{c+b}\right)$ is a negative binomial–distributed random variable. We can equivalently parameterize it as $y \sim \text{NB}(a, g(\zeta))$, where $g(\zeta) = 1 - \exp(-\zeta)$ is the Bernoulli–Poisson link [22] and $\zeta = \ln\left(1 + \frac{c}{b}\right)$.

**Property 3 (P3)**: if $y \sim \text{NB}(a, g(\zeta))$ and $l \sim \text{CRT}(y, a)$ is a Chinese restaurant table-distributed random variable, then $y$ and $l$ are equivalently jointly distributed as $y \sim \text{SumLog}(l, g(\zeta))$ and $l \sim \text{Pois}(a\zeta)$ [24].

### A.1  Forward-downward sampling

**Sampling transition matrix $\Pi^{(l)}$:**  The alternative model specification, with $\Theta$ marginalized out, assumes that $\left(Z_{1kt}^{(l)}, ..., Z_{K_l,k,t}^{(l)}\right) \sim \text{Multi}\left(x_{kt}^{(l+1,l)}, \left(\pi_{1k}^{(l)}, ..., \pi_{K_lk}^{(l)}\right)\right)$. Therefore, via the Dirichlet-multinomial conjugacy, we have

$$(\boldsymbol{\pi}_k^{(l)}|-) \sim \text{Dir}(\nu_1^{(l)}\nu_k^{(l)} + Z_{1k\cdot}^{(l)}, ..., \nu_{K_l}^{(l)}\nu_k^{(l)} + Z_{K_lk\cdot}^{(l)}) \ . \tag{14}$$

**Sampling loading factor matrix $\Phi^{(l)}$:** Given these latent counts, via the Dirichlet-multinomial conjugacy, we have

$$(\boldsymbol{\phi}_k^{(l)}|-) \sim \text{Dir}(\eta^{(l)} + A_{1k\cdot}^{(l)}, ..., \eta^{(l)} + A_{K_{l-1}k\cdot}^{(l)}) \ . \tag{15}$$

**Sampling $\delta_t^{(1)}$:** Via the gamma-Poisson conjugacy, we have

$$(\delta_t^{(1)}|-) \sim \text{Gam}\left(\varepsilon_0 + \sum_{v=1}^{V}x_{vt}^{(1)}, \varepsilon_0 + \sum_{k=1}^{K_1}\theta_{kt}^{(1)}\right), \text{ if } \delta_t^{(1)} \neq \delta_{t'}^{(1)} \text{ for } t \neq t'; \tag{16}$$

$$(\delta^{(1)}|-) \sim \text{Gam}\left(\varepsilon_0 + \sum_{t=1}^{T}\sum_{v=1}^{V}x_{vt}^{(1)}, \varepsilon_0 + \sum_{t=1}^{T}\sum_{k=1}^{K_1}\theta_{kt}^{(1)}\right), \text{ if } \delta_t^{(1)} = \delta^{(1)} \text{ for all } t. \tag{17}$$

**Sampling $\beta^{(l)}$:**

$$(\beta^{(l)}|-) \sim \text{Gam}\left(\varepsilon_0 + \gamma_0, \varepsilon_0 + \sum_{k=1}^{K_l}\nu_k^{(l)}\right) \tag{18}$$

**Sampling $v_k^{(l)}$ and $\xi^{(l)}$:**

$$(Z_{k1t}^{(l)}, ..., Z_{kK_lt}^{(l)}|-) \sim \text{Multi}\left(x_{kt}^{(l+1,l)}; \frac{\pi_{k1}^{(l)}\theta_{1,t-1}^{(l)}}{\sum_{k_l=1}^{K_l}\pi_{kk_l}^{(l)}\theta_{k_l,t-1}^{(l)}}, ..., \frac{\pi_{kK_l}^{(l)}\theta_{K_l,t-1}^{(l)}}{\sum_{k_l=1}^{K_l}\pi_{kk_l}^{(l)}\theta_{k_l,t-1}^{(l)}}\right), \tag{19}$$

To obtain closed-form conditional posterior for $v_k^{(l)}$ and $\xi^{(l)}$, we start with

$$(Z_{1k\cdot}^{(l)}, \cdots, Z_{kk\cdot}^{(l)}, \cdots, Z_{K_1k\cdot}^{(l)}) \sim \text{DirMult}(Z_{\cdot k\cdot}^{(l)}, (v_1^{(l)}v_k^{(l)}, \cdots, \xi^{(l)}v_k^{(l)}, \cdots, v_K^{(l)}v_k^{(l)})), \qquad (20)$$

where $Z_{k_1k\cdot}^{(l)} = \sum_{t=1}^{T} Z_{k_1kt}^{(l)}$ and $Z_{\cdot k\cdot}^{(l)} = \sum_{t=1}^{T} \sum_{k_1=1}^{K_l} Z_{k_1kt}^{(l)}$. Following Zhou [36], we draw a beta-distributed auxiliary variable:

$$(q_k^{(l)} \mid -) \sim \text{Beta}(Z_{\cdot k\cdot}^{(l)}, v_k^{(l)}(\xi^{(l)} + \sum_{k_1 \neq k} v_{k_1}^{(l)})). \qquad (21)$$

Consequently, we have

$$P(Z_{kk\cdot}^{(l)}, q_k^{(l)}) \propto \text{NB}(Z_{kk\cdot}^{(l)}; \xi^{(l)}v_k^{(l)}, q_k^{(l)}) \quad \text{and} \quad P(Z_{k_1k\cdot}^{(l)}, q_k^{(l)}) \propto \text{NB}(Z_{k_1k\cdot}^{(l)}; v_{k_1}^{(l)}v_k^{(l)}, q_k^{(l)}) \quad (22)$$

for $k_1 \neq k$. Next, we introduce the following auxiliary variables:

$$(h_{kk}^{(l)} \mid -) \sim \text{CRT}(Z_{kk\cdot}^{(l)}, \xi^{(l)}v_k^{(l)}) \quad \text{and} \quad (h_{k_1k}^{(l)} \mid -) \sim \text{CRT}(Z_{k_1k\cdot}^{(l)}, v_{k_1}^{(l)}v_k^{(l)}) \qquad (23)$$

for $k_1 \neq k$. We can then re-express the joint distribution over the variable in (22) and (23) as

$$Z_{kk\cdot}^{(l)} \sim \text{SumLog}(h_{kk}^{(l)}, q_k^{(l)}) \quad \text{and} \quad Z_{k_1k\cdot}^{(l)} \sim \text{SumLog}(h_{k_1k}^{(l)}, q_k^{(l)}) \qquad (24)$$

and

$$h_{kk}^{(l)} \sim \text{Pois}(-\xi^{(l)}v_k^{(l)}\ln(1-q_k^{(l)})) \quad \text{and} \quad h_{k_1k} \sim \text{Pois}(-v_{k_1}^{(l)}v_k^{(l)}\ln(1-q_k^{(l)})). \qquad (25)$$

Then, via the gamma-Poisson conjugacy, we have

$$(\xi^{(l)}\mid -) \sim \text{Gam}\left(\frac{\gamma_0}{K_l} + \sum_{k=1}^{K_l} h_{kk}^{(l)}, \ \beta^{(l)} - \sum_{k=1}^{K_l} v_k^{(l)}\ln(1-q_k^{(l)})\right). \qquad (26)$$

Note that when $l = L$ and $t = 1$, we have $\boldsymbol{\theta}_1^{(L)} \sim \text{Gam}\left(\tau_0 v_k^{(L)}, \tau_0\right)$ and $m_{k1}^{(L)} \sim$ Pois $\left(\tau_0(\zeta_2^{(L)} + \zeta_1^{(L-1)})\theta_{k1}^{(L)}\right)$, where $m_{k1}^{(1)} = A_{\cdot k1}^{(1)} + Z_{\cdot k2}^{(1)}$. So we can sample $(x_{k1}^{(L+1)} \mid -) \sim$ CRT$(m_{k1}^{(l)}, \tau_0 v_k^{(l)})$. Via **P3**, We can further get $x_{k1}^{(L+1)} \sim \text{Pois}(\zeta_1^{(L)}\tau_0 v_k^{(L)})$.

Next, because $x_k^{(L+1)}$ also depends on $v_k^{(L)}$, we introduce

$$n_k^{(l)} = h_{kk}^{(l)} + \sum_{k_1 \neq k} h_{k_1k}^{(l)} + \sum_{k_2 \neq k} h_{kk_2}^{(l)} \qquad (27)$$

for $l = 1, \ldots, L-1$ and

$$n_k^{(L)} = h_{kk}^{(L)} + \sum_{k_1 \neq k} h_{k_1k}^{(L)} + \sum_{k_2 \neq k} h_{kk_2}^{(L)} + x_{k1}^{(L+1)}. \qquad (28)$$

Then, via **P1**, we have

$$n_k^{(l)} \sim \text{Pois}(v_k^{(l)}\rho_k^{(l)}), \qquad (29)$$

where

$$\rho_k^{(l)} = -\ln(1-q_k^{(l)})(\xi^{(l)} + \sum_{k_1 \neq k} v_{k_1}^{(l)}) - \sum_{k_2 \neq k} \ln(1-q_{k_2}^{(l)})v_{k_2}^{(l)} \qquad (30)$$

for $l = 1, \ldots, L-1$ and

$$\rho_k^{(L)} = -\ln(1-q_k^{(L)})(\xi^{(L)} + \sum_{k_1 \neq k} v_{k_1}^{(L)}) - \sum_{k_2 \neq k} \ln(1-q_{k_2}^{(L)})v_{k_2}^{(L)} + \zeta^{(L)}\tau_0. \qquad (31)$$

Finally, via the gamma-Poisson conjugacy, we have

$$(v_k^{(l)}\mid -) \sim \text{Gam}\left(\frac{\gamma_0}{\beta^{(l)}} + n_k^{(l)}, \beta^{(l)} + \rho_k^{(l)}\right). \qquad (32)$$

---

**Algorithm 1** Backward-Upward–Forward-Downward Gibbs sampling for DPGDS

---
**for** iter = 1 : $B_L + C_L$ do Gibbs sampling **do**
  \ ⋆ *Collect local information*
  Backward-upward Gibbs sampling for $\{A_{vkt}^{(l)}\}_{v,k,t}$; $\{x_{kt}^{(l+1)}\}_{k,t}$; $\{x_{kt}^{(l+1,l)}\}_{k,t}$ ; $\{x_{kt}^{(l+1,l+1)}\}_{k,t}$;
  $\{Z_{k_1 k_2 t}^{(l)}\}_{k_1,k_2,t}$ with (8)-(11);
  Backward-upward calculating for $\{\zeta_t^{(l)}\}_t$;
  Forward-downward Gibbs sampling for $\{\boldsymbol{\theta}_t^{(l)}\}_t$ with (12);
  Sampling $\boldsymbol{\delta}^{(1)}$ with (16) or (17);
  \ ⋆ *Update global parameters*
  **for** $l = 1, 2, \cdots, L \quad and \quad k = 1, 2, \cdots, K_L$ **do**
    Update $\{\boldsymbol{\pi}_k^{(l)}\}_k$ from (14); Update $\{\boldsymbol{\phi}_k^{(l)}\}_k$ from (15); Update $\beta^{(l)}, \xi^{(l)}, \{\nu_k^{(l)}\}_k$ according to
    (26), (18), and (32);
  **end for**
**end for**

---

## A.2 SGMCMC for DPGDS

Although the Gibbs sampling algorithm for DPGDS has closed-form update equations discussed above, it requires handling all time-varying vectors in each iteration and hence has limited scalability [26]. To allow for tractable and scalable inference, in Section 3.3, we propose a SGMCMC method to infer the DGPDS using TLASGR-MCMC [27] to update $\{\boldsymbol{\Pi}^{(l)}\}_{l=1}^L$. In this section, we discuss how to update the other global parameters in detail, as described in Algorithm in 2.

**Sample the transmission matrix** $\{\boldsymbol{\Pi}^{(l)}\}_{l=1}^L$**:**

$$\left(\boldsymbol{\pi}_k^{(l)}\right)_{n+1} = \left[\left(\boldsymbol{\pi}_k^{(l)}\right)_n + \frac{\varepsilon_n}{M_k^{(l)}}\left[\left(\rho\tilde{\boldsymbol{z}}_{:k\cdot}^{(l)} + \boldsymbol{\eta}_{:k}^{(l)}\right) - \left(\rho\tilde{z}_{\cdot k\cdot}^{(l)} + \eta_{\cdot k}^{(l)}\right)\left(\boldsymbol{\pi}_k^{(l)}\right)_n\right]\right.$$
$$\left. + \mathcal{N}\left(0, \frac{2\varepsilon_n}{M_k^{(l)}}\left[\mathrm{diag}(\boldsymbol{\pi}_k^{(l)})_n - (\boldsymbol{\pi}_k^{(l)})_n(\boldsymbol{\pi}_k^{(l)})_n^T\right]\right)\right]_{\angle}. \tag{33}$$

**Sample the hierarchical topics** $\{\boldsymbol{\Phi}^{(l)}\}_{l=1}^L$**:** In DPGDS, the prior and likelihood of $\{\boldsymbol{\Phi}^{(l)}\}_{l=1}^L$ resemble those for $\{\boldsymbol{\Pi}^{(l)}\}_{l=1}^L$, so we also apply the TLASGR MCMC sampling algorithm on it as

$$\left(\boldsymbol{\phi}_k^{(l)}\right)_{n+1} = \left[\left(\boldsymbol{\phi}_k^{(l)}\right)_n + \frac{\varepsilon_n}{P_k^{(l)}}\left[\left(\rho\tilde{\boldsymbol{A}}_{:k\cdot}^{(l)} + \eta_0^{(l)}\right) - \left(\rho\tilde{A}_{\cdot k\cdot}^{(l)} + K_{l-1}\eta_0^{(l)}\right)\left(\boldsymbol{\phi}_k^{(l)}\right)_n\right]\right.$$
$$\left. + \mathcal{N}\left(0, \frac{2\varepsilon_n}{P_k^{(l)}}\left[\mathrm{diag}(\boldsymbol{\phi}_k^{(l)})_n - (\boldsymbol{\phi}_k^{(l)})_n(\boldsymbol{\phi}_k^{(l)})_n^T\right]\right)\right]_{\angle}, \tag{34}$$

where $M_k^{(l)}$ and $P_k^{(l)}$ are calculated using the estimated FIM, $\tilde{\boldsymbol{z}}_{:k\cdot}$, $\tilde{z}_{\cdot k\cdot}^{(l)}$, $\tilde{\boldsymbol{A}}_{:k\cdot}^{(l)}$, and $\tilde{A}_{\cdot k\cdot}^{(l)}$ come from the augmented latent counts $\mathbf{Z}^{(l)}$ and $\mathbf{A}^{(l)}$, $\boldsymbol{\eta}_{:k}^{(l)}$ and $\eta_0^{(l)}$ denote the prior of $\boldsymbol{\pi}_k^{(l)}$ and $\boldsymbol{\phi}_k^{(l)}$, and $[\cdot]_{\angle}$ denotes a simplex constraint; more details about TLASGR-MCMC for DLDA can be found in Cong et al. [27].

For other global variables, $\Lambda_g$, containing $\{\xi^{(l)}\}_{l=1}^L$ and $\{v_k^{(l)}\}_{l=1,k=1}^{L,K_l}$ (the hyper-parameter $\{\beta^{(l)}\}_{l=1}^L$ is set to 1 here), we find that it is enough to use a first-order SGMCMC method to sample them. Considering the efficiency and the performance, we use the stochastic gradient Nose-Hoover thermostat (SGNHT) to update all these variables, which has the potential advantage of making the system jump out of local models easier and reach the equilibrium state faster. Specifically, the dynamic system are defined by the following stochastic differential equations:

$$d\Lambda_g = \boldsymbol{p}dt, d\boldsymbol{p} = \boldsymbol{f}(\Lambda_g) - \tau\boldsymbol{p}dt + \sqrt{2A}\mathcal{N}(0, dt) \tag{35}$$
$$d\tau = (\frac{1}{n}\boldsymbol{p}^T\boldsymbol{p} - 1)dt \tag{36}$$

where $\boldsymbol{p}$ simulate the momenta in a system and $\tau$ is called the thermostat variable which ensures the system temperature to be constant. The stochastic force $\boldsymbol{f}(\Lambda_g) = -\nabla_{\Lambda_g} U(\Lambda_g)$, where $U(\Lambda_g)$ is the negative log-posterior of a Bayesian model, is calculated on a mini-batch subset of data or the other global parameters. Note that given the appropriate initial values of $\Lambda_g, \tau, \boldsymbol{p}, A$, it is only need to calculate the $\boldsymbol{f}(\Lambda_g)$ to update the $\Lambda_g$, which will be given.

**Calculate the stochastic force of $v_k^{(l)}$:**

$$U\left(\nu_k^{(l)}\right) = -\sum_{k=1}^{K_l} \log p\left(\boldsymbol{\pi}_k^{(l)}|\zeta^{(l)}, \nu_k^{(l)}\right) - \log p\left(\nu_k^{(l)}|\frac{\gamma_0}{K_l}, \beta^{(l)}\right), \tag{37}$$

$$\nabla_{\nu_k^{(l)}} U\left(\nu_k^{(l)}\right) = -\left[\sum_{k_1=1}^{K_l}\left(\nu_{k_1}^{(l)}\right)\log\left(\pi_{k_1 k}^{(l)}\right) + \sum_{k_2=1}^{K_l}\left(\nu_{k_2}^{(l)}\right)\log\left(\pi_{kk_2}^{(l)}\right) + \left(\zeta^{(l)} - 4\nu_k^{(l)}\right)\log \pi_{kk}^{(l)}\right]$$
$$- \frac{\left(\frac{\gamma_0}{K_l} - 1\right)}{\nu_k^{(l)}} + \beta^{(l)}. \tag{38}$$

**Calculate the stochastic force of $\xi^{(l)}$:**

$$U\left(\xi^{(l)}\right) = -\sum_{k=1}^{K_l} \log p\left(\pi_k^{(l)}|\xi^{(l)}\right) - \log p\left(\xi^{(l)}|\varepsilon_0, \varepsilon_0\right), \tag{39}$$

$$\nabla_{\xi^{(l)}} U\left(\xi^{(l)}\right) = -\sum_{k=1}^{K_l} \nu_k^{(l)} \log\left(\pi_{kk}^{(l)}\right) - \frac{(\varepsilon_0 - 1)}{\xi^{(l)}} + \varepsilon_0. \tag{40}$$

---

**Algorithm 2** Stochastic-gradient MCMC for DPGDS

---

Input: Data mini-batches; Output: Global parameters of DPGDS.

  **for** $i = 1, 2, \cdots$ **do**

    \ ⋆ *Collect local information*

    Backward-upward Gibbs sampling on the $i$th mini-batch for $\{A_{vkt}^{(l)}\}_{v,k,t}$; $\{x_{kt}^{(l+1)}\}_{k,t}$; $\{x_{kt}^{(l+1,l)}\}_{k,t}$ ; $\{x_{kt}^{(l+1,l+1)}\}_{k,t}$; $\{Z_{k_1 k_2 t}^{(l)}\}_{k_1,k_2,t}$ with (8)-(11);

    Backward-upward calculating for $\{\zeta_t^{(l)}\}_t$;

    Forward-downward Gibbs sampling for $\{\boldsymbol{\theta}_t^{(l)}\}_t$ with (12);

    Sampling $\boldsymbol{\delta}^{(1)}$ with (16) or (17);

    \ ⋆ *Update global parameters*

    **for** $l = 1, 2, \cdots, L$   *and*   $k = 1, 2, \cdots, K_L$ **do**

      Update $M_k^{(l)}$ according to Cong et al. [27], and then $\{\phi_k^{(l)}\}_k$ with (34); Update $M_k^{(l)}$ according to [27], and then $\{\pi_k^{(l)}\}_k$ with (33);

    **end for**

    Update $\xi^{(l)}$, $\{\nu_k^{(l)}\}_k$, and $\beta^{(l)}$ with SGNHT [20]

  **end for**

---

# B Results on Bouncing ball

In Fig. 7, we show the original data and the one-step prediction frames of five different algorithms. The frames in each subplot is arranged by time from left to right and top to bottom. We find that the most difficult prediction is the frames that describe how the balls move after the collision, such as observing the fourth row and ninth row. We find that comparing with the original data, a good model means that two balls can be separated soon after the collision, while a bad model means that two balls have unreasonable trajectories. According to this action mechanism, we can see that DPGDS outperforms the others.

Figure 7: (a) The original data and the one-step prediction results on the bouncing ball date set by (b) TSBN, (c) PGDS, (d) TSBN-4, (e) DTSBN, (f) DPGDS, and .

# C  Results on ICEWS 2007-2009

In order to understand the DPGDS better, based on the results inferred on ICEWS 2007-2009 via a three-hidden-layer DPGDS, with the size of 200-100-50, we show in Fig. 8 how some example topics are hierarchically and temporally related to each other, and how their corresponding latent representations evolve over time. Similar findings and conclusions can be reached according to Fig. 8 like ICEWS 2001-2003 in Figs. 5 and 6. In Fig. 9, we also present a subset of the transition matrix $\mathbf{\Pi}^{(l)}$ in each layer, corresponding to the top ten topics, some of which have been displayed in Fig. 8.

(a)

(b)

Figure 8: Topics and their temporal trajectories inferred by a three-layer DPGDS from the ICEWS 2007-2009 dataset. (a) Some example topics that are hierarchically or temporally related; (b) The temporal trajectories of some inferred latent topics.

Figure 9: Learned transition structure on ICEWS 2007-2009 from the same DPGDS depicted in Fig. 8. Shown in (a)-(c) are transition matrices for layers 1, 2 and 3, respectively, with a darker color indicating a larger transition weight (between 0 and 1).

# D Results on GDELT 2015-2018

To add more empirical study on scalability, we have collected GDELT data from February 2015 to July 2018 (temporal granularity of 15 mins), resulting in a count matrix with $V = 1000$ and $T \approx 120,000$. For such a long time series, Backward-Upward–Forward-Downward Gibbs sampling for DPGDS is impractical to run as a single iteration takes nearly 3000 seconds. GPDM is trained with a batch algorithm, which is also too time consuming to run for this dataset. However, by taking short sequences at random locations from the data, we can run both DTSBN [4] and the proposed DPGDS using SGMCMC. Here, we use $[K^{(1)}, K^{(2)}, K^{(3)}] = [200, 100, 50]$ for both DPGDS and DTSBN and choose the length of each short sequence to be $T = 60$. As shown in Fig. 10, we present how DTSBN and the proposed DPGDS progress over time, evaluated with MP, MR and PP. It takes about $6000s$ for DTSBN and DPGDS-SGMCMC to converge. Clearly, our DPGDS-SGMCMC is scalable and clearly outperforms DTSBN.

Figure 10: Shown in (a)-(c) are MP, MR, PP, respectively, as the function of time for GDELT 2015 -2018.