[Reviews · NeurIPS 2018]

Reviewer 1



This paper presents a deep Poisson-gamma dynamical systems (DPGDS) to model sequentially observed multivariate count data, which is built on top of PGDS and PGBN. It can be seen as: each time stamp is associated with PGBN. The corresponding latent layers $\theta$ are connected with a transition matrix. From figure 1, one can tell each column is GBN and each row is also a GBN, which, I think, is the major contribution of this paper. Given the construction, the inference algorithm should be straightforward by following PGDS and PGBN. The experimental work is quite comprehensive. The proposed model was compared with the model in the same vein. It would be good to add a bit more empirical study on the scalability of the proposed model, given that Stochastic MCMC is used. In general, it is an ineresting paper.

Reviewer 2



This paper generalizes the Poisson-gamma dynamical systems (PGDS) to more effectively mine hierarchical latent structures in multivariate time series. The resulting model is called Deep PGDS, which places a hierarchical structure on the latent factors, specified again by a Gamma transition equation The closed-form Gibbs sampling algorithm is developed with the same techniques that exploit the relationship between Poisson/Multinomial, Poisson/NB, etc., as well as the stochastic gradient MCMC version. Overall, the presentation of this paper is very good, considering its notation-heavy nature, and the technical part is solid as well as its experimental section. I quite enjoyed reading the paper, nevertheless, my main concerns are: 1) on the algorithmic side, aside from the algebraic derivation, it's unclear about the technical contributions given that the techniques used are standard; 2) with the complicated MCMC inference algorithm, how practical it is for large-scale datasets?

Reviewer 3



This paper presents Deep Poisson-Gamma Dynamical System (DPGDS) for modeling temporal multivariate count data. It is based on previously developed Gamma-Belief networks, extended to the dynamical scenarios by adding transitions of latent units in consecutive times. The paper is well written, and connections to previous papers are explained clearly. Major comments: 1. While the temporal structure is based on transition of latent units in a Markov manner, the authors' claim about better capturing long-range temporal changes should be justified more clearly. 2. As the shape parameter in the Gamma distributions corresponding to latent units is summation of two terms, does imposing Dirichlet distributions on \Phi and \Pi makes the latent units completely identifiable? 3. I suggest to compare your performance with dynamical methods based on Gaussian Process Latent Variable Models, which are incorporated specifically to have more flexible temporal structure, if possible. Minor comments: 1. In line 30, "separated" should be replaced by "separately". 2. The sentence starting at line 46 is better to be rewritten more clearly.